

# A comparison of two common sample preparation techniques for lipid and fatty acid analysis in three different coral morphotypes reveals quantitative and qualitative differences

Jessica A. Conlan[1,2], Melissa M. Rocker[3] and David S. Francis[1,2]

[1] School of Life and Environmental Sciences, Deakin University, Warrnambool, Victoria, Australia
[2] Australian Institute of Marine Science, Townsville, Queensland, Australia
[3] School of Life and Environmental Sciences, Deakin University, Geelong, Victoria, Australia

## ABSTRACT

Lipids are involved in a host of biochemical and physiological processes in corals. Therefore, changes in lipid composition reflect changes in the ecology, nutrition, and health of corals. As such, accurate lipid extraction, quantification, and identification is critical to obtain comprehensive insight into a coral's condition. However, discrepancies exist in sample preparation methodology globally, and it is currently unknown whether these techniques generate analogous results. This study compared the two most common sample preparation techniques for lipid analysis in corals: (1) tissue isolation by air-spraying and (2) crushing the coral *in toto*. Samples derived from each preparation technique were subsequently analysed to quantify lipids and their constituent classes and fatty acids in four common, scleractinian coral species representing three distinct morphotypes (*Acropora millepora*, *Montipora crassotuberculata*, *Porites cylindrica*, and *Pocillopora damicornis*). Results revealed substantial amounts of organic material, including lipids, retained in the skeletons of all species following air-spraying, causing a marked underestimation of total lipid concentration using this method. Moreover, lipid class and fatty acid compositions between the denuded skeleton and sprayed tissue were substantially different. In particular, the majority of the total triacylglycerol and total fatty acid concentrations were retained in the skeleton (55–69% and 56–64%, respectively). As such, the isolated, sprayed tissue cannot serve as a reliable proxy for lipid quantification or identification in the coral holobiont. The *in toto* crushing method is therefore recommended for coral sample preparation prior to lipid analysis to capture the lipid profile of the entire holobiont, permitting accurate diagnoses of coral condition.

# INTRODUCTION

Coral reefs worldwide are declining at an alarming pace due to the increasing diversity, frequency, and scale of human impacts (*Lesser, 2012*; *Hughes et al., 2017*). Consequently,

Corresponding author
Jessica A. Conlan,
conlan@deakin.edu.au,
jessconlan@live.com.au

the urgency and prevalence of internationally-integrated research supporting coral reef management, rehabilitation, and aquaculture efforts has increased (*Leal et al., 2016*).

Scleractinian corals are commonly used as biomonitors for phenological and ecological phenomena in tropical reef ecosystems, given their high sensitivity to physical and chemical changes in the marine environment (*Goffredo et al., 2011*; *Filho et al., 2012*). Specifically, coral lipid reserves serve as a universal proxy for coral health status given their ubiquitous nature (*Anthony et al., 2009*; *Lesser, 2012*). Lipids are a major component of the coral proximate composition (10–40% of dry biomass) and their constituent classes and fatty acids play important roles in energy storage, cell membrane structure, and overall fitness (*Bergé & Barnathan, 2005*; *Farre, Cuif & Dauphin, 2010*). The quantity and nature of coral lipids vary significantly in response to environmental factors such as season and food availability, as well as with physiological processes such as photosynthesis, respiration, and reproduction (*Arai et al., 1993*; *Imbs, 2013*). As such, lipid analysis is an important and prevalent aspect of coral biology, with over 560 publications relating to coral lipids published since 1970 (ISI Web of Science).

Likewise, coral lipid investigations are gaining momentum as tools for understanding current and future climate change impacts on coral reefs, since total lipid content has been shown to determine a coral's ability to offset the adverse responses to climate change-associated stressors (*Baumann et al., 2014*; *Towle, Enochs & Langdon, 2015*). For example, large lipid stores can mitigate growth reductions caused by reduced photosynthate transfer as well as greater energy demands on calcification in response to ocean warming and acidification (*Baumann et al., 2014*; *Towle, Enochs & Langdon, 2015*). Additionally, the size of a coral's lipid stores have been shown to significantly influence the onset timing of severe bleaching and subsequent mortality. Indeed, under high bleaching rates, corals with 'full' initial lipid stores have been shown to survive for twice as long as corals with half-depleted lipid stores (*Anthony et al., 2009*). Moreover, examining the corals' lipid class and fatty acid composition in addition to the total lipid content evidences key metabolic changes elicited by bleaching, and thus determine a corals' ability to initially resist bleaching, followed by the timing and capacity for full recovery (*Rodrigues, Grottoli & Pease, 2008*).

Therefore, accurate quantification and subsequent identification of coral lipids and their constituents is of significant importance to their reliability as biomarkers of coral health and their capacity to predict the timing and severity of major stress events associated with climate change. However, within the literature, discrepancies exist between sample preparation techniques prior to lipid analysis, and it is currently unknown whether these techniques generate analogous results.

The two most common techniques for coral sample preparation prior to lipid analyses are air-spraying and *in toto* crushing. The first, air-spraying, is a modified version of the Water-Pik method described by *Johannes & Wiebe (1970)*. This method involves completely removing the tissue from the coral skeleton using an airgun connected to a source of compressed air inside a thick, polyethylene bag (*Deschaseaux et al., 2013*). The sprayed tissue is collected in the bag and combined with filtered seawater to form a slurry that is then homogenised, frozen, and freeze-dried to obtain the tissue alone. This tissue isolate is then extracted for lipids, excluding the denuded skeleton (*Szmant & Gassman,*

*1990*). Air-spraying presents a flexible option, since the resultant isolates can be used for further biometrics such as zooxanthellae counting (tissue) (*Edmunds & Gates, 2002*) and surface area measurements (skeleton) (*Veal et al., 2010*). This technique also ensures that the samples are maximally utilised, since conservative coral sampling is often necessary to limit environmental impacts on donor reefs and reduce associated collection expenses. As such, performing lipid analyses on the isolated coral tissue alone remains prevalent in coral biochemical research (e.g., *Seemann et al., 2013*; *Towle, Enochs & Langdon, 2015*; *Crandall et al., 2016*; *Lim, Bachok & Hii, 2017*).

The second method, *in toto* crushing, utilises the coral as a whole; crushing the intact skeleton and tissue while frozen within a mortar and pestle, and using the resultant powder for lipid analyses (*Saunders et al., 2005*; *Deschaseaux et al., 2013*). Crushing takes into consideration the presence of the 'skeletal organic material' (SOM) that is integrated throughout the skeleton. This includes the skeletal organic matrix, which controls skeletal growth by inducing or inhibiting biomineral crystal nucleation (*Allemand et al., 1998*; *Farre, Cuif & Dauphin, 2010*; *Hemond, Kaluziak & Vollmer, 2014*), the gastrovascular system (*Parrin et al., 2010*), and other organic material such as entrapped tissue and zooxanthellae (*Tambutté et al., 2007*). Importantly, SOM in the skeleton has been shown to differ biochemically from the tissue (*Dauphin, Cuif & Massard, 2006*; *Ramos-Silva et al., 2014*). However, comprehensive quantification and identification of lipids isolated in the tissue and those retained by the skeleton does not exist. Thus, it is not yet known whether isolated tissue obtained with the air-spraying technique accurately represents the lipid quantity and composition of the whole coral. This reduces the reliability of lipids as a biomarker of several aspects of coral health, including reproductive status, growth, nutritional integrity, response to environmental change, and overall fitness, potentially leading to inaccurate diagnoses of coral condition. This, in turn, can lead to inadequate or inappropriate decisions relating to reef management, jeopardising the success of coral monitoring, rehabilitation, and aquaculture efforts. Furthermore, discrepancies between sample preparation techniques limits the ability for direct comparison between studies, reducing the efficiency of global investigations.

Here, we explore and compare the performance of these two sample preparation methods to determine their effectiveness for accurate lipid, lipid class, and fatty acid (FA) analyses of the coral holobiont. In order to show the applicability of each method for different coral morphotypes, we tested four common coral genera with three different skeletal and tissue morphologies: *Acropora millepora* and *Porites cylindrica;* representing perforate, branching species, *Montipora crassotuberculata;* representing perforate, plate forming species, and *Pocillopora damicornis;* representing imperforate, branching species. Where differences between the two methods were detected, their causes and ramifications are discussed.

## MATERIALS AND METHODS

### Sample collection

Forty coral nubbins (∼length 5 cm) from each of four species of common scleractinian genera; *Acropora millepora*, *Montipora crassotuberculata*, *Porites cylindrica*, and *Pocillopora*

*damicornis*, were collected from the Great Barrier Reef, Queensland, Australia between the 18–19th of April 2015 (three genotypes species$^{-1}$) (Field collections were approved by the Great Barrier Reef Marine Park Authority: G12/35236.1). *M. crassotuberculata* were collected from Pelorus Island (lat.: 18°548′S, long.: 146°504′E). All other species were collected from Davies Reef from two sites (lat.: 18°496′S, long.: 147°376′E and lat.: 18°499′S, long.: 147°379′E). Corals were then transferred to the National Sea Simulator facility at The Australian Institute of Marine Science (AIMS, Townsville, Australia, lat.: 16°177′S, long.: 145°271′E).

## Sample preparation

All nubbins were weighed prior to sample preparation. For each species, 20 nubbins were prepared with the air-spraying method as described by *Szmant & Gassman (1990)*. Tissue was removed from the coral skeleton using a jet of high-pressure air from a hand gun (80 psi, ∼1 cm distance to coral). All sprayed tissue was captured in a thick, polyethylene bag containing 10 ml of ultrafiltered seawater (0.04 µm filtration). Nubbins were sprayed scrupulously for 10 min, ensuring that all surface tissue was removed, as confirmed visually by the completely white surface of the denuded skeletons. To further ensure that no surface tissue remained, the denuded skeletons were triple-rinsed in ultrafiltered seawater, which was also collected. The tissue slurry was then poured from the plastic bag into a falcon tube, and the bag was double-rinsed with ultrafiltered seawater that was also collected. The slurry was then homogenised for 20 s (Ultra-Turrax T10B; IKA Labortechnik, Staufen im Breisgau, Germany). The isolated tissue and denuded skeletons were then re-weighed separately in order to quantify their individual proximate parameters, and individual contribution to the total composition. The denuded skeletons were kept and subjected to all analyses alongside the isolated tissue and *in toto* samples. Following this, the sprayed tissue and denuded skeleton results were recombined *ex post facto* in order to account for any discrepancies detected between the air-spraying method (isolated tissue only) and the *in toto* crushing method.

Between each sample, all apparatus were thoroughly cleaned with methanol ($CH_3OH$) and rinsed with seawater. The remaining 20 nubbins from each species were left untreated and these, along with the tissue slurry and denuded skeletons, were frozen at $-20\ °C$. All samples were then freeze-dried for 96 h (Labconco FreeZone, Kansas City, MO, USA), removing all moisture from the tissue slurry. Following freeze-drying, both the denuded and intact skeleton samples were placed inside a stainless steel mortar and pestle (cleaned with methanol), which was placed inside a manual laboratory hydraulic press (Model C; Fred S. Carver Inc., Summit, NJ, USA), and pressurised to 70 kN, crushing the corals to a fine powder.

## Proximate analysis
### Total lipid and ash

The tissue isolate, denuded skeleton, and intact samples were extracted for total lipid content according to the method described in *Conlan et al. (2014)*. Dry samples were soaked overnight in 3 mL of dichloromethane: methanol ($CH_2Cl_2$:$CH_3OH$). The following morning, this mixture was filtered and the solid residue re-suspended and further soaked

for 10 min with 3 mL of $CH_2Cl_2$:$CH_3OH$, followed by a further filtration step. This process was repeated three times. The combined filtrates (∼9 mL) were then combined with 4.5 mL of KCl (0.44%) in $H_2O$/$CH_3OH$ (3:1, v/v), shaken vigorously and settled overnight. The following morning, the bottom layer containing the extracted lipid was recovered (∼2.5 mL) and the solvent was evaporated under nitrogen. The lipid content was then quantified gravimetrically on a 4-figure balance (AB 204; Mettler-Toledo, Greifensee, Switzerland).

Total ash was determined by incineration in a muffle furnace (Model WIT; C & L Fetlow, Blackburn, Victoria, Australia) at 450 °C for 12 h. The ash content was subtracted from the total composition to obtain ash-free dry weight (AFDW), which excludes the inorganic component.

### Lipid class composition

Lipid class analysis was determined using an Iatroscan MK 6 s thin layer chromatography-flame ionisation detector (Mitsubishi Chemical Medience, Tokyo, Japan) according to the method of *Conlan et al. (2014)*. Each sample was spotted in duplicate on silica gel S5-chromarods (5 μm particle size) with lipid separation following a two-step elution sequence: (1) elution of phosphatidyletanolamine (PE), phosphatidylserine and phosphatidylinositol (PS-PI), phosphatidylcholine (PC), and lysophosphatidylchloline (LPC) was achieved in a $CH_2Cl_2$:$CH_3OH$:$H_2O$ (50:20:2, by volume) solvent system run to half height (∼15 min); and (2) after air drying, elution of wax esters (WAX), triacylglycerol (TAG), free fatty acid (FFA), 1,2-diacylglycerol (1,2DAG), and sterol (ST) was achieved in a $C_{16}H_{14}$:$(C_2H_5)_2O$:$CH_2O_2$ (60:15:1.5, by volume) solvent system run to full height (∼30 min). Since glycolipids commonly elute with monoacylglycerols and pigments, the term "acetone mobile polar lipid" (AMPL) was used (*Parrish, Bodennec & Gentien, 1996*). AMPL was quantified using the 1-monopalmitoyl glycerol standard (Sigma-Aldrich Co., USA), which has demonstrated a response that is intermediate between glycoglycerolipids and pigments (*Parrish, Bodennec & Gentien, 1996*).

### Fatty acid and fatty alcohol composition

*Acid catalysed methylation.* Following extraction, FA were esterified into methyl esters using the acid catalysed methylation method (*Christie, 2003*). 100 μL of 23:0 (0.75 mg mL$^{-1}$) was added as an internal standard (Sigma-Aldrich, Inc., St. Louis, MO, USA) alongside 2 mL of freshly prepared $CH_3COCl$: $CH_3OH$ (1:10, v/v). Sample vials were then shaken and placed in an oven at 100 °C for 1 h. Once cool, 2 mL of $K_2CO_3$ (1 M) and 1.7 mL of $C_{16}H_{14}$ were added and the sample centrifuged. The $C_{16}H_{14}$ supernatant was then recovered and placed in a vial for subsequent gas chromatograph (GC) injection.

*Gas chromatography.* FA methyl esters were identified using an Agilent Technologies 7890A GC System (Agilent Technologies; Santa Clara, CA, USA) equipped with a BPX70 capillary column (120 m × 0.25 mm internal diameter, 0.25 μm film thickness, SGE Analytical Science, Ringwood, VIC, Australia), a flame ionization detector (FID), an Agilent Technologies 7693 auto sampler, and a splitless injection system. The injection volume was 1 μ L and the injector and detector temperatures were 300 °C and 270 °C, respectively. The temperature program was 60 °C held for 2 min, then from 60 °C to

150 °C at 20 °C min$^{-1}$, and held at 150 °C for 2 min, then from 150 °C to 205 °C at 1.5 °C min$^{-1}$, then from 205 °C to 240 °C at 5 °C min$^{-1}$, and held at 240 °C for 24 min. The carrier gas was helium at a constant flow of 1.5 mL min$^{-1}$. Each FA was identified relative to known external standards (Sigma-Aldrich, Inc., St. Louis, MO, USA and Nu-Chek Prep Inc., Elysian, MN, USA), using GC ChemStation (Rev B.04.03; Agilent Technologies; Santa Clara, CA, USA). The resulting peaks were then corrected by theoretical relative FID response factors (*Ackman, 2002*) and quantified relative to the internal standard.

## Statistical analysis

Untransformed data were analysed statistically using RStudio (*R Studio Team, 2015*; *R Development Core Team, 2016*). For between-group comparisons (tissue vs skeleton), a Welch Two-Sample $t$-test was used, at a 0.05 significance level. Although data were non-normal and slightly skewed to the right, the $t$-test was considered robust given the moderate sample size ($n = 20$) (*Lumley et al., 2002*). Principal component analysis (PCA) was also performed to reduce and group 20 individual FA (expressed as % total FA) to more concisely explain and visualise overall variation (*Kassambara & Mundt, 2016*). PCA ellipses show 95% confidence intervals. Graphs were prepared using the ggplot2 package (*Wickham, 2009*).

## RESULTS

### Proximate composition

Based on dry weight, the intact crushing method showed that the large majority of the holobiont consisted of ash in all species (∼944–954 mg g dry sample$^{-1}$), while the organic fraction constituted only a small portion (∼46.2–66.1 mg g dry sample$^{-1}$). *A. millepora* contained the highest total lipid concentration (9.93 ± 0.64 mg g sample$^{-1}$), while *P. damicornis* recorded the lowest (2.74 ± 0.22 mg g sample$^{-1}$) (Table 1).

Recombining the sprayed tissue and denuded skeleton results from the air-spraying method *ex post facto* revealed significant loss of tissue, and consequently lipid, when compared to the intact samples, with *A. millepora*, *M. crassotuberculata*, and *P. damicornis* recording a loss of ∼one third, and *P. cylindrica* ∼one quarter of the total lipid. Conversely, no difference was detected in the ash content between the two samples.

Comparing the two isolates alone showed, unsurprisingly, that the denuded skeleton accounted for the large majority of the total sample (∼960–989 mg g sample$^{-1}$) (Table 1). The majority of the total organic material was retained in the denuded skeleton, and this was quantitatively similar across all genera (∼33.9–36.6 mg g sample$^{-1}$) (Table 1). The relative contribution of the skeleton to the total organic fraction was highest for *P. damicornis* (∼83.6%), and lowest for *A. millepora* (∼63.2%) (Table 1). On the other hand, the concentrations of organic material in the tissue varied between genera, with *A. millepora* recording the highest (21.3 ± 2.20 mg g sample$^{-1}$) (∼37% of total), and *P. damicornis* the lowest (6.72 ± 1.30 mg g sample$^{-1}$) (∼16% of total).

The relative contribution of total lipid from each isolate was remarkably consistent between *A. millepora*, *M. crassotuberculata*, and *P. damicornis* (∼42% skeleton, ∼58% tissue), although the quantitative amounts differed (∼0.76–2.62 mg g sample$^{-1}$ skeleton,

**Table 1  Proximate composition of four scleractinian species using air-spraying and *in toto* crushing sample preparation techniques.** Intact, samples prepared with *in toto* crushing method; Recombined, combined results of isolated skeleton and tissue prepared with the air-spraying method; Skeleton, denuded skeleton samples; Tissue, isolated tissue samples.

| Species | (mg g sample$^{-1}$) | Crush | Air-spray | | |
| --- | --- | --- | --- | --- | --- |
| | | Intact | Recombined | Skeleton | Tissue |
| *A. millepora* | Total sample | 1,000 ± 0.00 | 1,000 ± 0.00 | 960 ± 1.80[a] | 40.1 ± 1.80[a] |
| | Ash | 944 ± 2.30 | 943 ± 2.20 | 923 ± 2.20[a] | 19.8 ± 2.20[a] |
| | Organic | 66.1 ± 2.30 | 57.1 ± 2.20 | 36.3 ± 2.20[a] | 21.3 ± 2.20[a] |
| | Lipid | 9.93 ± 0.64[*] | 6.20 ± 0.61[*] | 2.62 ± 0.61[a] | 3.58 ± 0.61[a] |
| | Lipid (mg g AFDW$^{-1}$) | 178 ± 10.6[*] | 110 ± 11.6[*] | 47.5 ± 11.6[a] | 62.5 ± 11.6[a] |
| *M. crassotuberculata* | Total sample | 1,000 ± 0.00 | 1,000 ± 0.00 | 967 ± 2.10[a] | 33.3 ± 2.10[a] |
| | Ash | 947 ± 2.30 | 950 ± 2.50 | 934 ± 2.50[a] | 16.2 ± 2.50[a] |
| | Organic | 53.2 ± 2.30 | 50.4 ± 2.20 | 33.9 ± 2.20[a] | 16.1 ± 2.20[a] |
| | Lipid | 6.42 ± 0.52[*] | 4.30 ± 0.50[*] | 1.82 ± 0.50 | 2.48 ± 0.50 |
| | Lipid (mg g AFDW$^{-1}$) | 122 ± 10.3[*] | 87.7 ± 9.80[*] | 38.1 ± 9.80 | 49.6 ± 9.80 |
| *P. cylindrica* | Total sample | 1,000 ± 0.00 | 1,000 ± 0.00 | 989 ± 0.80[a] | 11.2 ± 0.80[a] |
| | Ash | 949 ± 2.40 | 951 ± 1.80 | 947 ± 1.80[a] | 3.80 ± 1.80[a] |
| | Organic | 51.1 ± 2.30 | 49.4 ± 2.20 | 36.6 ± 1.80[a] | 12.4 ± 1.80[a] |
| | Lipid | 3.44 ± 0.21[*] | 2.5 ± 0.33[*] | 1.26 ± 0.33 | 1.24 ± 0.33 |
| | Lipid (mg g AFDW$^{-1}$) | 67.8 ± 5.20 | 51.1 ± 6.50 | 26.3 ± 6.50 | 24.8 ± 6.50 |
| *P. damicornis* | Total sample | 1,000 ± 0.00 | 1,000 ± 0.00 | 979 ± 0.80[a] | 21.1 ± 0.80[a] |
| | Ash | 954 ± 2.60 | 959 ± 1.30 | 950 ± 1.30[a] | 8.63 ± 1.30[a] |
| | Organic | 46.2 ± 2.30 | 41.1 ± 2.20 | 34.3 ± 1.30[a] | 6.72 ± 1.30[a] |
| | Lipid | 2.74 ± 0.22[*] | 1.8 ± 0.24[*] | 0.76 ± 0.24 | 1.04 ± 0.24 |
| | Lipid (mg g AFDW$^{-1}$) | 61.3 ± 4.30[*] | 45.1 ± 5.00[*] | 19.6 ± 5.00 | 25.5 ± 5.00 |

**Notes.**

Values are presented as means ± SEM ($n = 20$).

[*]indicate significant differences between intact and recombined samples ($P < 0.05$).

[a]indicate significant differences between denuded skeletons and isolated tissues ($P < 0.05$).

and ∼1.04–3.58 mg g sample$^{-1}$ tissue). Meanwhile, *P. cylindrica* contained almost equal amounts of the total lipid concentration in the skeleton and tissue (1.26 ± 0.30 and 1.24 ± 0.30 mg g sample$^{-1}$, respectively).

### Lipid class composition
#### Intact vs. recombined samples
The lipid class composition of the intact samples showed the storage component of all species to be dominated by TAG, followed by WAX (Table 2). *P. cylindrica* was an exception, exhibiting comparatively low TAG concentrations. This translated to the total storage lipid for this species (307 ± 23.9 mg g lipid$^{-1}$), which was low compared to the other species (∼375–433 mg g lipid$^{-1}$). For all genera, the polar lipids were dominated by AMPL, PE, PS-PI, and PC.

Recombining the sprayed tissue and denuded skeleton results from the air-spraying method *ex post facto* showed some discrepancies when compared to the intact samples for all species. In particular, there were significantly higher concentrations of FFA and lower PE and LPC in the intact samples compared to the recombined for

**Table 2  Lipid class composition of intact and recombined samples of four scleractinian species prepared using air-spraying and *in toto* crushing sample preparation techniques.**  Intact, samples prepared with crushing method; Recombined, combined results of isolated skeleton and tissue prepared with the air-spraying method.

| (mg g lipid$^{-1}$) | A. millepora | | M. crassotuberculata | | P. cylindrica | | P. damicornis | |
|---|---|---|---|---|---|---|---|---|
| | Intact | Recombined | Intact | Recombined | Intact | Recombined | Intact | Recombined |
| WAX | 123 ± 6.70 | 112 ± 6.39 | 85.9 ± 4.79 | 104 ± 22.4 | 88.1 ± 2.60 | 98.1 ± 5.05 | 95.6 ± 2.92 | 104 ± 7.20 |
| TAG | 194 ± 32.9 | 172 ± 32.3 | 170 ± 33.0 | 143 ± 29.6 | 70.7 ± 5.18 | 75.2 ± 8.78 | 189 ± 25.9 | 205 ± 32.6 |
| FFA | 41.8 ± 3.17[*] | 27.0 ± 1.99[*] | 21.5 ± 1.90[*] | 32.9 ± 2.81[*] | 36.1 ± 10.6 | 28.0 ± 3.05 | 45.8 ± 3.04[*] | 30.1 ± 3.49[*] |
| 1,2 DAG | 74.2 ± 7.57 | 82.1 ± 7.37 | 97.1 ± 5.27 | 83.3 ± 10.4 | 112 ± 2.06 | 109 ± 11.4 | 82.2 ± 7.95 | 76.8 ± 10.2 |
| ST | 80.1 ± 4.38 | 71.7 ± 4.09 | 74.7 ± 3.89 | 77 ± 4.81 | 79.2 ± 4.31 | 86.3 ± 5.52 | 71.7 ± 2.84 | 80.8 ± 6.87 |
| AMPL | 175 ± 11.0 | 184 ± 10.9 | 173 ± 15.6 | 210 ± 16.2 | 171 ± 6.63 | 185 ± 9.82 | 134 ± 5.62[*] | 166 ± 11.4[*] |
| PE | 84.4 ± 7.02[*] | 106 ± 6.41[*] | 118 ± 10.4 | 114 ± 6.25 | 120 ± 1.67 | 133 ± 9.6 | 99.0 ± 3.85 | 110 ± 7.96 |
| PSPI | 104 ± 14.8 | 102 ± 14.1 | 115 ± 18.6 | 96.7 ± 17.0 | 153 ± 13.4[*] | 104 ± 18.3[*] | 122 ± 14.6 | 109 ± 13.6 |
| PC | 123 ± 7.49 | 135 ± 7.27 | 133 ± 8.79 | 126 ± 8.07 | 151 ± 2.85 | 142 ± 13.0 | 123 ± 5.97 | 105 ± 10.2 |
| LPC | nd[*] | 8.97 ± 4.21[*] | 12.1 ± 6.52 | 12.6 ± 8.58 | 19.1 ± 8.75 | 39.4 ± 8.52 | 38.1 ± 9.51[*] | 14.1 ± 6.72[*] |
| ∑STORAGE | 433 ± 32.8 | 393 ± 33.9 | 375 ± 34.2 | 363 ± 33.5 | 307 ± 9.76 | 310 ± 17.1 | 412 ± 29.0 | 415 ± 28.3 |
| ∑STRUCTURAL | 567 ± 32.8 | 607 ± 33.9 | 625 ± 34.2 | 637 ± 33.5 | 693 ± 9.76 | 690 ± 17.1 | 588 ± 29.0 | 585 ± 28.3 |

**Notes.**

Values are presented as means ± SEM ($n = 20$).

nd, not detected.

*indicate significant differences between intact and recombined samples within each species ($P < 0.05$).

*A. millepora*, significantly lower FFA in the intact samples compared to the recombined for *M. crassotuberculata*, significantly higher PSPI in the intact compared to the recombined for *P. cylindrica*, and significantly lower AMPL and higher LPC in the intact compared to the recombined for *P. damicornis* ($P < 0.05$).

### *Tissue vs. skeleton—relative contribution*

*A. millepora*, *M. crassotuberculta*, and *P. damicornis* contained significantly higher proportions of ST in the tissue ($P < 0.05$) (Fig. 1). These species also contained higher levels of AMPL and all individual phospholipids in the tissue, however, the former was only significant for *A. millepora* and *M. crassotuberculata,* and the latter only for *A. millepora* ($P < 0.05$). Meanwhile, *P. cylindrica* contained significantly higher amounts of ST in the skeleton compared to the tissue ($P < 0.05$). For all genera, the majority of WAX and TAG occurred in the skeleton, and the latter was significant for *A. millepora* and *P. damicornis* ($P < 0.05$).

### Fatty acid and fatty alcohol composition
#### *Intact vs. recombined samples*

The total FA concentration was highest in *A. millepora* and *P. cylindrica* (33.4 ± 9.42 and 34.9 ± 8.52% lipid, respectively) (Table 3), followed by *M. crassotuberculata* (30.0 ± 9.78% lipid), while *P. damicornis* contained the lowest (25.8 ± 10.3% lipid). All genera were dominated by SFA (~50% FA), followed by PUFA (~20–30% FA), and MUFA (~11–15% FA). The dominant SFA was 16:0 for all species (~25–38% FA), while 18:1n-9 was the major MUFA for all species except *M. crassotuberculata*, which contained similar amounts of

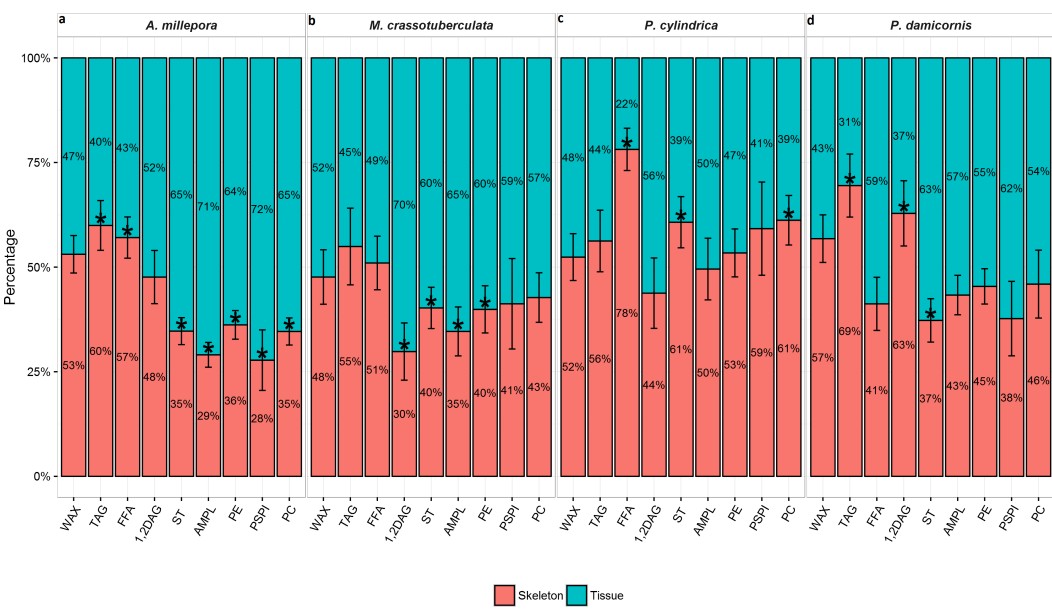

**Figure 1** **Lipid class composition of denuded skeleton and isolated tissue of four scleractinian species prepared with the air-spraying method—relative contribution (% total) Values are presented as means ± SEM ($n = 20$).** * denote significant differences between stacked bars within each species (tissue vs skeleton) ($P < 0.05$).

20:1n-11. The dominant PUFA were species-specific, being 20:5n-3 (EPA) for *A. millepora*, 22:4n-6 for *M. crassotuberculata* and *P. cylindrica*, and 22:6n-3 (DHA) for *P. damicornis*.

Due to the quantitative loss in total lipid content, recombining the sprayed tissue and denuded skeleton results from the air-spraying method *ex post facto* showed correspondingly lower amounts of all individual FA compared to the intact samples. The total FA concentration (% lipid) was significantly higher in the intact samples compared to the recombined for all species except *P. damicornis* ($P < 0.05$). Furthermore, all species contained significantly higher proportions of PUFA in the intact samples compared to the recombined (% FA). On the other hand, all species contained higher proportions of SFA in the recombined samples compared to the intact, and this was statistically significant for *A. millepora* and *P. cylindrica* ($P < 0.05$). The individual FA, EPA and DHA exhibited higher concentrations in the intact samples compared to the recombined, and the former was statistically significant for all species except *M. crassotuberculata*, while the latter was significant for *A. millepora* and *P. cylindrica* ($P < 0.05$).

### Tissue vs. skeleton—relative contribution

The total FA (% total) proportion was greatest in the skeleton for all genera (~56–64%), and was significant compared to the tissue for *P. cylindrica* and *P. damicornis* ($P < 0.05$) (Fig. 2). Generally, the major FA groups were dispersed uniformly between the skeleton and tissue. However, for *A. millepora*, there were significantly higher proportions of PUFA in the tissue (~55%), while *P. cylindrica* contained a higher proportion of PUFA in the skeleton (~56%) ($P < 0.05$).

**Table 3** **Fatty acid and fatty alcohol composition of intact and recombined samples of four scleractinian species prepared using air-spraying and *in toto* crushing sample preparation techniques.** Intact, samples prepared with crushing method; Recombined, combined results of isolated skeleton and tissue prepared with the air-spraying method

| (% fatty acids) | A. millepora | | M. crassotuberculata | | P. cylindrica | | P. damicornis | |
|---|---|---|---|---|---|---|---|---|
| | Intact | Recombined | Intact | Recombined | Intact | Recombined | Intact | Recombined |
| 16 Ac | 0.42 ± 0.03 | 0.47 ± 0.03 | 0.21 ± 0.03 | 0.18 ± 0.03 | 0.19 ± 0.03 | 0.18 ± 0.04 | 0.38 ± 0.02 | 0.40 ± 0.04 |
| DMA 18:0 | 1.53 ± 0.16 | 1.20 ± 0.14 | 0.20 ± 0.02 | 0.23 ± 0.04 | 2.67 ± 0.21 | 2.88 ± 0.24 | 2.00 ± 0.19 | 2.26 ± 0.36 |
| 16:OH | 5.68 ± 0.80 | 6.06 ± 1.04 | 2.23 ± 0.53 | 2.02 ± 0.67 | 0.88 ± 0.35 | 0.38 ± 0.18 | 2.77 ± 0.38 | 2.83 ± 0.58 |
| ∑Fatty Alcohol | 7.77 ± 0.62 | 8.07 ± 0.88 | 2.65 ± 0.53 | 2.57 ± 0.64 | 3.79 ± 0.42 | 4.22 ± 0.31 | 5.63 ± 0.37 | 6.02 ± 0.48 |
| 14:0 | 3.55 ± 0.20 | 3.72 ± 0.18 | 3.38 ± 0.22 | 3.71 ± 0.30 | 1.97 ± 0.26 | 2.17 ± 0.37 | 3.19 ± 0.18 | 3.60 ± 0.19 |
| 16:0 | 29.9 ± 1.78 | 33.7 ± 1.38 | 34.2 ± 2.70 | 37.8 ± 2.77 | 24.2 ± 0.78* | 27.3 ± 1.10* | 32.2 ± 0.83 | 31.7 ± 1.88 |
| 18:0 | 11.6 ± 0.53 | 12.4 ± 0.76 | 8.92 ± 0.34 | 9.65 ± 0.29 | 10.5 ± 0.69 | 12.2 ± 0.46 | 13.8 ± 0.47* | 16.2 ± 0.53* |
| ∑SFA | 48.5 ± 1.24* | 53.5 ± 0.75* | 51.9 ± 2.42 | 56.4 ± 2.22 | 42.4 ± 1.35* | 47.9 ± 1.42* | 52.7 ± 0.94 | 55.8 ± 1.43 |
| 18:1n-9 | 3.92 ± 0.32 | 4.70 ± 0.24 | 3.06 ± 0.16* | 5.26 ± 0.51* | 5.98 ± 0.21* | 7.60 ± 0.24* | 5.98 ± 0.32* | 7.08 ± 0.38* |
| 20:1n-11 | 3.85 ± 0.31 | 3.26 ± 0.22 | 3.85 ± 0.50 | 3.79 ± 0.43 | 4.19 ± 0.39* | 2.97 ± 0.20* | 1.99 ± 0.11 | 1.75 ± 0.12 |
| ∑MUFA | 12.6 ± 0.18 | 12.5 ± 0.24 | 11.0 ± 0.38* | 14.6 ± 1.00* | 14.3 ± 1.13 | 13.6 ± 0.58 | 15.6 ± 0.45 | 16.6 ± 0.79 |
| 20:5n-3 | 11.1 ± 0.93* | 8.31 ± 0.64* | 4.25 ± 0.47 | 3.11 ± 0.35 | 8.96 ± 0.22* | 7.03 ± 0.33* | 3.83 ± 0.28* | 2.57 ± 0.31* |
| 22:6n-3 | 3.75 ± 0.08* | 3.12 ± 0.08* | 2.49 ± 0.17 | 2.07 ± 0.14 | 7.50 ± 0.33* | 5.71 ± 0.35* | 7.12 ± 0.16 | 6.34 ± 0.44 |
| 18:3n-6 | 3.55 ± 0.19 | 3.50 ± 0.19 | 4.21 ± 0.19 | 3.69 ± 0.24 | 1.58 ± 0.14 | 1.53 ± 0.12 | 1.32 ± 0.04 | 1.28 ± 0.09 |
| 20:4n-6 | 4.53 ± 0.52 | 3.60 ± 0.41 | 7.26 ± 0.57* | 4.68 ± 0.47* | 7.43 ± 0.40 | 6.91 ± 0.57 | 4.38 ± 0.42* | 2.28 ± 0.32* |
| 22:4n-6 | 3.16 ± 0.36 | 2.62 ± 0.31 | 9.83 ± 1.21 | 7.00 ± 1.14 | 7.45 ± 0.31 | 7.01 ± 0.53 | 3.20 ± 0.31 | 2.97 ± 0.40 |
| ∑PUFA | 31.2 ± 1.86* | 26.0 ± 1.38* | 34.4 ± 2.60* | 26.5 ± 2.21* | 39.5 ± 0.88* | 34.3 ± 1.52* | 26 ± 0.97* | 21.5 ± 0.82* |
| ∑n-3 PUFA | 17.6 ± 1.14* | 13.7 ± 0.76* | 8.53 ± 0.81 | 6.59 ± 0.65 | 19.6 ± 0.65* | 14.8 ± 0.63* | 12.6 ± 0.35* | 10.3 ± 0.51* |
| ∑n-6 PUFA | 13.4 ± 0.71 | 12.2 ± 0.61 | 25.4 ± 1.79* | 19.5 ± 1.63* | 19.5 ± 0.98 | 19.5 ± 1.29 | 13.3 ± 0.71* | 11.1 ± 0.59* |
| Total[a] (% lipid) | 34.9 ± 2.84* | 26.1 ± 2.21* | 30.0 ± 3.36* | 18.7 ± 2.06* | 33.4 ± 3.14* | 16.4 ± 0.78* | 25.8 ± 3.10 | 29.2 ± 2.16 |

**Notes.**

Values are presented as means ± SEM ($n = 20$).

16 Ac, Hexadecyl acetate; DMA 18:0, 1,1-dimethoxyoctadecane; 16:OH, 1-hexadecanol.

*indicate significant differences between intact and recombined samples within each species ($P < 0.05$).

[a]Total value also includes the minor fatty acids: 8:0, 10:0, 12:0, 15:0, 17:0, 20:0, 21:0, 22:0, 24:0, 14:1n-5, 15:1n-5, 16:1n-7, 17:1n-7, 18:1n-7, 18:1n-7trans, 18:1n-9trans, 20:1n-9, 22:1n-9, 24:1n-9, 16:2n-4, 16:3n-4, 18:3n-4, 18:3n-3, 18:4n-3, 20:4n-3, 22:3n-3, 22:5n-3, 24:6n-3, 18:2n-6, 18:2n-6trans, 20:2n-6, 20:3n-6, 22:2n-6, 22:5n-6.

### Principal components

Both groups were separated fairly well along the first two principal components, which described a large proportion of the data (*A. millepora*: 76.4%, *M. crassotuberculata*: 54.0%, *P. damicornis*: 64.0%, *P. cylindrica*: 60.1%) (Fig. 3). For all species, 16:0 was a major contributor to the separation of the skeleton from the tissue, being largely retained by the skeleton (Fig. 4). The fatty alcohol, 16:OH, along with 21:0, 20: $1n - 11$, and EPA, were more strongly associated with the tissue for all species. The major contributor to the second principal component was DHA for *A. millepora*, *P. cylindrica*, and *P. damicornis*.

## DISCUSSION

The present study evaluated the comparative efficacy of two sample preparation methods of scleractinian corals for accurate lipid and FA analysis. Lipids and their constituent classes and FA play important roles in coral energy storage, cell membrane structure, and overall fitness (*Bergé & Barnathan, 2005*; *Farre, Cuif & Dauphin, 2010*). As such, lipid analysis is

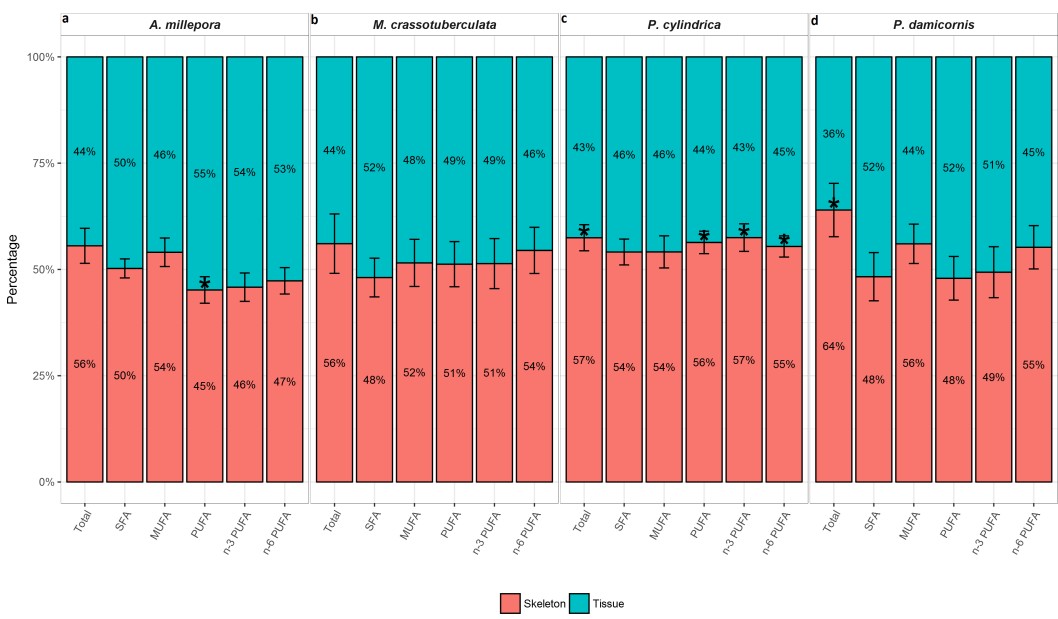

**Figure 2** Fatty acid composition of denuded skeleton and isolated tissue of four scleractinian species prepared with the air-spraying method—relative contribution (% total) Values are presented as means ± SEM (*n* = 20). * denote significant differences between stacked bars within each species (tissue vs skeleton) (*P* < 0.05).

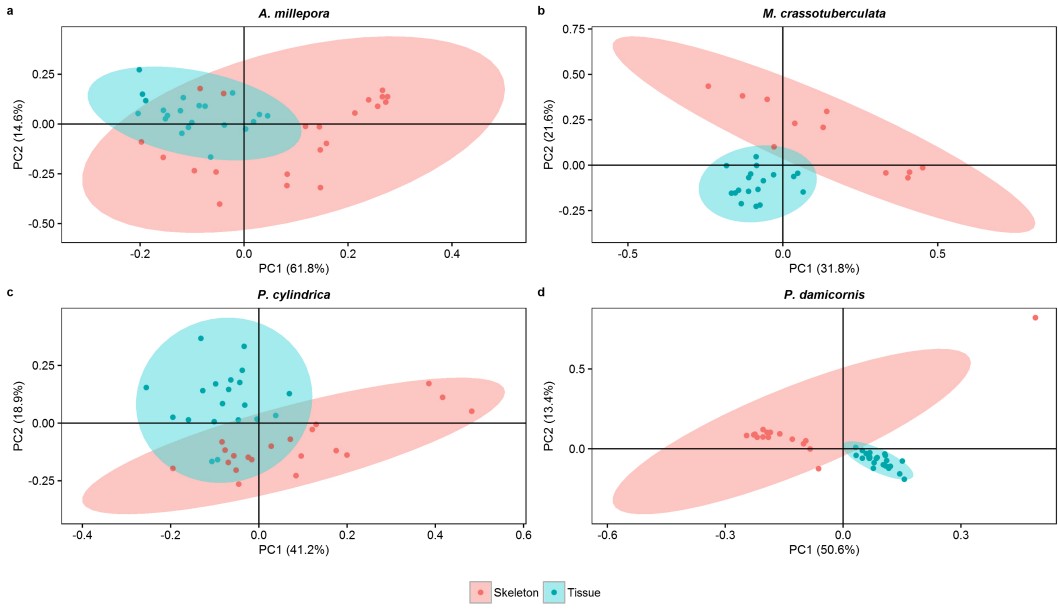

**Figure 3** Score plot of principal component analysis of fatty acid and fatty alcohol profiles (based on % fatty acids) of denuded skeleton and isolated tissue of four scleractinian species prepared with the air-spraying method (ellipses show 95% confidence intervals).
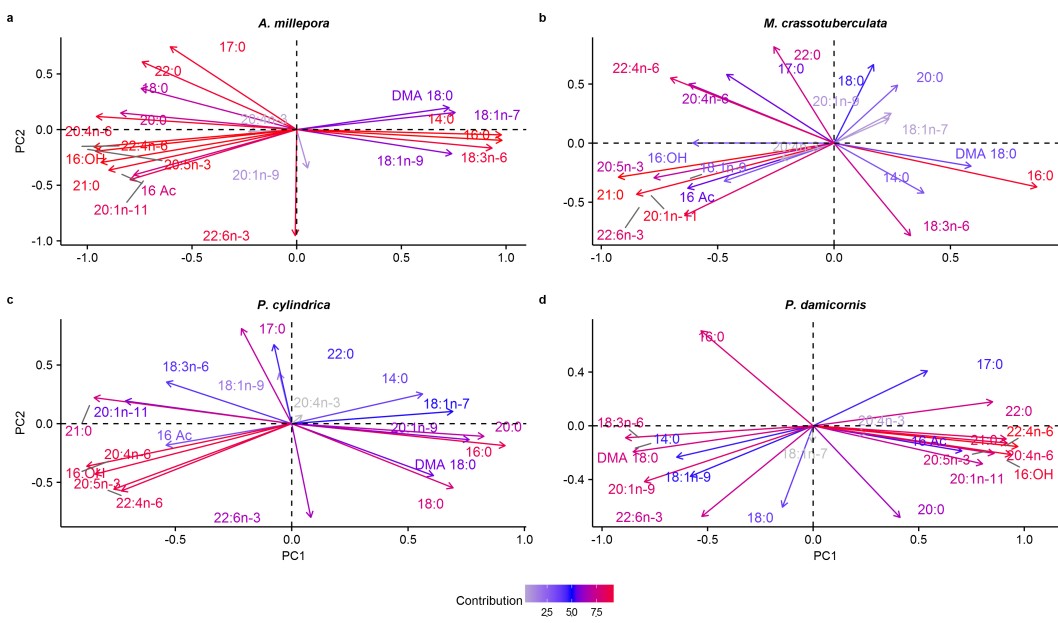

**Figure 4** **Principal component analysis loading plot of fatty acid and fatty alcohol profiles of denuded skeleton and isolated tissue of four scleractinian species prepared with the air-spraying method (% fatty acids).** Colour gradient shows percentage contribution to overall variance. Fatty alcohol abbreviations: 16 Ac, Hexadecyl acetate; DMA, 18:0, 1,1-dimethoxyoctadecane; 16:OH, 1-hexadecanol.

an important and prevalent aspect of coral biology, and the use of accurate and ubiquitous methodology in coral biochemistry ensures that subsequent actions in coral monitoring, rehabilitation, and aquaculture efforts are appropriate and well-informed.

Recombining the sprayed tissue and denuded skeleton *ex post facto* and comparing with the intact samples revealed significant tissue and hence lipid loss across all samples prepared with the air-spraying technique (Table 1). This loss likely occurred during the additional steps required by this method, increasing the risk of tissue loss through residue on apparatus, including the airgun nozzle, polyethylene bag, and homogeniser shaft, despite thorough rinsing. Additionally, the increased handling time increases the risk of lipid oxidation through excessive air exposure, potentially altering the lipid profile. Lipid oxidation is also a major drawback of some alternative, rarefied methods of coral lipid extraction. For example, the process of decalcifying the coral skeleton with an acetic acid solution (e.g., *Yamashiro et al., 1999*; *Rodríguez-Troncoso et al., 2011*) is likely to elicit major alteration of the lipid profile, since acetic acid has been shown to cause significant lipid oxidation and fatty acid hydrolysis (*Sajiki, Takahashi & Takahashi, 1995*). Additionally, the gravimetric calculation method of soaking the oven-dried coral *in toto* in an organic solvent and re-weighing the re-dried skeleton to quantify total lipids (e.g., *Ward, 1995*; *Pisapia, Anderson & Pratchett, 2014*) involves excessive sample handling and exposure to heat - greatly increasing the risk of lipid degradation and rendering this technique unusable for qualitative analyses. In contrast, the *in toto* crushing method used in the present study requires far less sample handling and laboratory equipment, preserving the integrity of the

lipid profile. Moreover, intact skeletons are only processed once freeze-dried and re-frozen, minimising the risk of oxidation and making residue easy to recover.

In addition to lipid loss, the results conclusively demonstrate that the large majority of the coral's total organic fraction is retained in the skeleton for all genera, including almost half the total lipid concentration (Table 1). This is despite scrupulously air-spraying the skeletons for ten minutes until they were uniformly white, and triple-rinsing the sprayed skeletons in seawater in an attempt to completely denude the skeletons of organic material. As such, excluding the denuded skeleton and analysing the isolated tissue alone for proximate composition incurs a significant underestimation of the total organic content and subsequent lipid concentration of the coral holobiont. Moreover, since lipid only constituted 2.5–8.7% of the total organic material (Table 1), the majority of the organic material recorded in both the skeleton and tissue went unclassified. This remaining portion likely consists of protein, amino acids, and carbohydrates, which are prevalent in a coral's organic fraction (Dauphin & Cuif, 1997; Allemand et al., 1998). Clearly, the total concentration of these compounds would also be severely underestimated using the air-spraying method. However, these represent critical nutrients in coral life functions since protein fuels tissue growth and calcification, while carbohydrates represent a quickly-mobilised energy source that sustain coral metabolism (Oku et al., 2002; Ramos-Silva et al., 2014). As such, this method can be deemed inappropriate not only for lipid analyses, but all proximate analyses.

The relative partitioning of structural complexity is different among the four morphotypes investigated. In particular, the genera, Acropora, Montipora, and Porites grow perforate skeletons with a relatively simple external structure, yet great internal complexity (Dauphin, Cuif & Massard, 2006; Farre, Cuif & Dauphin, 2010; Work & Aeby, 2010; Yost et al., 2013). The internal complexity of perforate corals is largely comprised of the intricate gastrovascular system, which penetrates deep into the skeleton and forms a three-dimensional environment for the dispersion of symbionts and organic matter (Hii, Soo & Liew, 2008; Veal et al., 2010; Work & Aeby, 2010; Davy, Allemand & Weis, 2012). In some cases, pockets of skeletal tissue are completely confined by the skeleton, and this internal tissue network cannot realistically be evacuated through the use of air-spraying.

On the other hand, the Pocillopora genus is known to produce imperforate skeletons, possessing only a thin layer of external tissue (Yost et al., 2013). Therefore, the persistence of >80% of the total organic material in the P. damicornis skeleton, including 42% of the total lipids, was surprising. However, this is likely attributable to two factors: Firstly, branching imperforate corals such as P. damicornis exhibit high levels of external structural complexity, including immersed corallites (Veron, 2000), compared to perforate genera (Yost et al., 2013). This external structural complexity impedes the ability of air-spraying to completely denude the skeleton, despite the rigorous and standardised application of the method.

Residual surface tissue has previously been demonstrated using the Water-Pik method (see Johannes & Wiebe, 1970) in both perforate (Bachok, Mfilinge & Tsuchiya, 2006) and imperforate (Brahmi et al., 2012) coral skeletons. Additionally, the Water-Pik method does not entirely remove thick, fibrous mesoglea, including zooxanthellae, in several

coral genera, including *Montipora*, *Pocillopora,* and *Porites* (*Johannes & Wiebe, 1970*). Presumably, this also applies to the air-spraying method, as both rely on pressurised water or air driving tissue from the skeleton. Yet the lipid content has been shown to be highest in the mesoglea and zooxanthellae, as well as the lower half of polyps, which contain the gastrodermis (*Al-Sofyani, 1994*). This is supported by coral tissue histology, which shows high amounts of lipid droplets in the gastrodermal layer of *P. damicornis*, with the epidermal layer containing far less (*Luo et al., 2009*; *Kopp et al., 2015*).

Secondly, persistent organic material in the denuded skeleton likely includes the organic matrix, for which lipids have been shown to play a significant role in forming (*Marin & Luquet, 2008*; *Adamiano et al., 2014*). Changes in production rates of the organic matrix has been attributed to growth and skeletal deposition (*Stolarski, 2003*). Furthermore, dynamic changes in lipid compositions of the host gastrodermis are known to reflect the endosymbiotic status (*Luo et al., 2009*). Since these represent significant processes in coral biology, inclusion of the skeletal component in lipid analyses is imperative in order to gain an accurate insight into a coral's physiological and biochemical condition.

The persistent lipid in the denuded skeletons also differed in chemical nature from the sprayed tissue. Storage lipids (WAX, TAG, FFA, and 1,2-DAG) are generally associated with energy supply, while structural lipids (AMPL, PE, PS-PI, and PC) are important for the membrane lipid bilayer, and cell membrane stability (*Lee, Hagen & Kattner, 2006*). Generally, the storage lipids were found in higher concentrations in the skeleton, while the structural lipids predominated in the tissue for all species except *P. cylindrica* (Fig. 1). The prevalence of storage lipids in the skeleton may be ascribable to a higher proportion of host tissue, including the gastrodermis, persisting deeper into the skeleton, as up to 90% of storage lipid has been found to reside in the host tissue rather than the zooxanthellae, largely in the form of WAX and TAG (*Imbs, Yakovleva & Pham, 2010*; *Chen et al., 2015*). Underestimation of storage lipids in corals has ramifications for biological studies, as WAX and TAG are considered to be the most important lipid species with respect to energetic status, which describes the amount of available energy compared to the energy required (*Anthony et al., 2009*; *Imbs, 2013*). In particular, concentrations of storage lipids are known to alter in response to coral metabolic requirements (*Oku et al., 2002*), reproduction (*Arai et al., 1993*), and zooxanthellae activity (*Oku et al., 2002*), which are key processes in coral biology and ecology.

Structural differences extended to the FA composition, which showed marked variation between the denuded skeleton and sprayed tissue, likely reflecting the larger proportion of host tissue in the denuded skeleton and zooxanthellae in the sprayed tissue, as well as functional specialisation of internal and external tissues (*Imbs, Yakovleva & Pham, 2010*; *Chen et al., 2015*). While zooxanthellae densities were not quantified in this work, high zooxanthellae contents have been previously recorded in isolated tissue using the Water-Pik method (*Edmunds & Gates, 2002*). Moreover, although zooxanthellae have been shown to reside within the gastrovascular system (*Domart-Coulon et al., 2006*), this is largely in the upper portion to gain access to light (*Goldberg, 2002*), and may thus be more readily removed through air-spraying.

Higher total FA concentrations were evident in the skeleton for all species (although this was only statistically significant for *P. cylindrica* and *P. damicornis*), which is reflective of the different lipid class compositions between the two isolates; the skeleton contained higher levels of TAG, which contains three esterified FA, while the sprayed tissue was richer in phospholipids, ST, and AMPL, which possess two or less esterified FA (*Lee, Hagen & Kattner, 2006*).

Despite a relatively uniform distribution of the major FA classes, SFA, MUFA, and PUFA, between the skeleton and tissue, the individual FA differed substantially between the two isolates. This was illustrated in the separation achieved by PCA between the denuded skeleton and sprayed tissue, and correlates with the findings of *Chen et al. (2015)*; that FA moieties of each lipid species differ between host and symbiont tissues.

For example, the FA 14:0, 16:0, and 20:0 contributed largely to the separation of the skeleton from the tissue for most species (Fig. 4), and these generally predominate in host tissues compared to zooxanthellae (*Treignier et al., 2008*). This is also consistent with the higher abundance of WAX and TAG in the skeleton, since SFA, particularly 16:0, are generally the major FA moieties of these classes in cnidarians (*Yamashiro et al., 1999*). Furthermore, n-6 PUFA were present in higher proportions in the denuded skeleton compared to the sprayed tissue for *M. crassotuberculata*, *P. cylindrica*, and *P. damicornis* (Fig. 2), and these have previously been shown to predominate in coral host tissue (*Treignier et al., 2008*).

The lipid and FA profile of the sprayed tissue may also be specialised to cope with external changes and threats, such as environmental conditions and disease. For instance, 16:OH, 21:0, 20:1n-11, and EPA generally showed stronger associations with the tissue for most species. The fatty alcohol, 16:OH, is recognised as an inhibitor of bacterial fouling in some coral species (*Dobretsov et al., 2015*), while EPA provides immune function (*Bergé & Barnathan, 2005*), as well as photo-protection from ultraviolet radiation (*Pilkington et al., 2011*). Furthermore, higher levels of phospholipids in the tissue compared to the skeleton for *A. millepora*, *M. crassotuberculata*, and *P. damicornis* may reflect the necessity of the outer tissues to cope with seasonal fluctuations in temperature, since regulation of phospholipid composition is suggested to reflect thermal tolerance in corals (*Revel et al., 2016*).

These results clearly demonstrate compartmentalisation of lipid classes and FA in the internal and external tissues of the coral holobiont. As such, this study conclusively demonstrates the inadequacy of the air-spraying technique to provide accurate identification and subsequent quantification of the total lipid and FA profile of corals, and this extends to all proximate analyses. Use of the intact crushing method for coral biochemical analyses is therefore recommended, since it is robust against tissue loss and accounts for the entire chemical composition of the holobiont.

Should studies require additional biometrics rendered impossible by the *in toto* crushing method, such as zooxanthellae densities and surface area, the use of separate, replicate samples to accommodate these analyses is recommended.

## ACKNOWLEDGEMENTS

The authors thank the SeaSim team at The Australian Institute of Marine Science (AIMS) and Deakin University's School of Life and Environmental Sciences staff for technical assistance throughout the project.

### Funding

The authors received no funding for this work.

### Competing Interests

The authors declare there are no competing interests.

### Author Contributions

- Jessica A. Conlan conceived and designed the experiments, performed the experiments, analyzed the data, wrote the paper, prepared figures and/or tables, reviewed drafts of the paper.
- Melissa M. Rocker and David S. Francis conceived and designed the experiments, wrote the paper, reviewed drafts of the paper.

### Field Study Permissions

The following information was supplied relating to field study approvals (i.e., approving body and any reference numbers):

Field collections were approved by the Great Barrier Reef Marine Park Authority.

### Data Availability

The raw data has been provided as Supplemental Files.

### Supplemental Information

Supplemental information for this article can be found online at http://dx.doi.org/10.7717/peerj.3645#supplemental-information.

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
