# Peer review of "A comparison of two common sample preparation techniques for lipid and fatty acid analysis in three different coral morphotypes reveals quantitative and qualitative differences"

_PeerJ, doi:10.7717/peerj.3645_

## Round 0.1 · original submission · Minor Revisions

Please take fully into account the comments of both the referees before resubmitting.

Reviewer 1 ·

Basic reporting

The work deals with the comparison of two methods of sample preparation prior to lipid extraction and analysis in corals: crushing the whole sample, against blowing apart the soft tissues. Although the answer to the question of the effectivity of both methods seems obvious in that the recovery should be better through the processing the whole sample, the paper shows elegantly how the analysis of the soft tissue alone produces a clear qualitative and quantitative underestimation of lipids, lipid classes and fatty acids, and how these components are partitioned between the skeleton and the tissue. The work is correctly written, easy to read and follow, well introduced and the methodology exposed adequately. The results are clear and generally well discussed except for certain points that I will raise below. Of particular concern is the drawing of conclusions based on differences not strictly statistically significant. I would suggest to add a table with the fatty acid composition similar to the one for lipid classes in the supplementary material. This would give a clear picture of the fatty acid profiles and even abbreviations (fatty alcohols, DMA…..), otherwise the reader has to go through a complex set of raw data to get this information.

Experimental design

In general terms I have little to say about the experimental design except for some particular points that I have preferred to outline in next section

Validity of the findings

Points that need attention:
Title: Is the topic of the work an analytical error in strict terms? Would not it be better to talk about bias or underestimation? Perhaps methodological error?
Legends: although the color codes are ok, consider the same sequence for all the legends (tissue skeleton in Fig.1 and skeleton tissue in Fig. 2). Use the same abbreviation for Sterol in text and figures. Explain that * denote significant differences between stacked bars (see below).
L 210: please explain the between-group comparison. Is it tissue-skeleton? Add explanation to figure legends.
Is it possible that through the whole sample crushing method, external lipids (i.e. fouling) are being analysed? Consider adding this to the discussion.
L213: how are the ellipses drawn? The point in the upper right corner of the score plot for P. damicornis in Fig.4 may be an outlier. Is it forcing the shape and confidence interval of the ellipse of the skeleton scores to go far beyond the actual distribution of the scores?
L219-L220: Unify the number of decimals for ash and organic fractions.
L220: although all samples were freeze-dried (L145) I should suggest to specify “dry weight”
L253: AMPL is not higher in P. damicornis. PS-PI and PC are not significantly higher for M. crassotuberculata!!!!!!! unless asterisks are missing in the Fig. Only statistical significance should account for real differences, and more in such a variable set of data.
L306: about the discussion of sample deterioration caused by lipid oxidation. No mention to the addition of antioxidants in the extraction method is made. I would suggest to provide an explanation for this and add a few lines here commenting the issue.
L392: as above, this is a too generalist assessment. A simple look to Figs. 2 and 3 would suffice to note that not all the differences invoked are significant (they are only for P. cylindrical and P. damicornis), and thus are not differences.
L408: Not for M. crassotuberculata.
L412: EPA does not seem to be associated with the tissue of P. cylindrical.
Final sentence L431: given the structural variability discussed in the text, and the incomplete extraction of soft tissues obtained through air-spraying, would not it be too risky to suggest these correction factors?

Re-check references… Symbiodinium should be in italics in refs Revel et al and Yost et al.

Additional comments

Please consider the changes and suggestions outined above. Although I think that the mayor discussion and conclusion lines of the work are correct, care should be taken about the discussion and conclusions drawn from differences not statistically justified.

Reviewer 2 ·

Basic reporting

This manuscript presents interesting results on the lipid composition of 4 common coral species evaluated using 2 different preparation techniques. This topic is in the scope if the journal.
The manuscript is well written and structure conforms to PeersJ standards. The raw data are supplied so the treatment of the data can be checked.

The introduction gives a good background on the subject and this study is highly relevant due to lack of data on lipid composition of the different parts of coral and the role of lipids in coral biology.

I found that the title did not reflect exactly the content of the manuscript. Authors characterise the lipid composition of 4 common coral species using 2 sample preparation techniques.

Figure 1 can be removed as same data are presented in table 1 in mg/g sample and it is easy to calculate the proportion. I think that table 2 should be included in the manuscript and not as supplementary material. Figure 2: it is not easy to read, would be better to have these data presented in a table like supplementary table 2. The comparison between whole coral/tissue and skeleton would be easier.

The discussion could be reduced in length and better structured, in a more logical order. There is a mix between comparison of skeleton vs tissue lipid composition, comparison of sample preparation technique, the role of the different lipid fractions in coral biology and the consequences of methods used for lipid determination. In its present form it is quite confuse (should be discussed in link with the aim of the study (see my comments on that).

Lines 311-318: not clear for me

The number of references cited can be reduced (71 references), keeping only the most relevant.

Experimental design

I found that the aim of the study could be more clearly explained at the end of the introduction because there are 2 different aspects treated in this paper: 1) quantification and identification of lipids extracted from skeleton and tissue (not studied before); 2) comparison of lipid composition between 2 preparation methods “air-spraying” (analysis conducted only on tissue) and “in toto crushing” (analysis on whole coral).

All; the techniques used to determine lipid and fatty acid composition have been described in details in Colan et al 2014, so this part of material and methods can be reduced. Lipids were extracted on all samples based on Folch method, slightly modified.

I’ve one question regarding the sample preparation (lines 142-143): why did you use water to rinse the skeleton instead of a solvent mix?

Line 166 lipids were quantified by gravimetric method.

Line 212: FA are expressed as percent of total FA

Validity of the findings

New results on lipid composition of coral are provided in this study and data set is robust and valid.
More details on the statistical treatment should be given (normality of the data, transformation for %).

Discussion is supported by the data presented.
Just one comment on Lines 439-444: to apply and recommend the use of a correction factor it is necessary to analyse more samples with the 2 preparation methods. You cannot recommend a correction factor with only 20 samples analysed.

Additional comments

This is an interesting study and the results obtained are worth publishing.
However the manuscript can be reduced in length and the discussion better structured.

Reviewer 3 ·

Basic reporting

.

Experimental design

.

Validity of the findings

.

Additional comments

The manuscript “A comparison of two common sample preparation techniques for lipid and fatty
acid analysis in corals reveals analytical error in three different morphotypes” by Conlan et al.
examines two lipid extraction techniques commonly applied to coral samples, finding that the in toto
method gave a more complete lipid profile than the air-spraying technique.

The manuscript is generally well written and presented and, while no major issues, I have included a
few comments for the authors to take into consideration.

Introduction

The introduction, while presented well, is perhaps a little bit lengthy. Although of some relevance,
Lines 42-70 could perhaps be condensed into one paragraph without losing the overall background
to the study.

Materials and Methods

The methodology is sound and is easy for a reader to follow and replicate. However, for the
quantification of lipids how were the air-sprayed samples calculated? For example, were intact
nubbins weighed first and the denuded skeleton weighed following the removal of coral tissue so
that the tissue and denuded skeleton lipid parameters could be quantified? (authors allude to this in
the discussion, Lines 303-306). Similarly, are results based on a dry or wet basis – please indicate.

Line 135: Change “1cm” to “1 cm”
Line 136: Change “0.04μm” to “0.04 μm”
Line 160: Change to “(3:1, v/v)”
Line 163: Was the total lipid made up to a known concentration, which solvents were used?
Line 189: Change to “(1:10, v/v)
Lines 174, 176, 188: Elsewhere the empirical formulas of chemicals are used but here they are
written – please be consistent throughout the text.

Results

Results are generally well presented and described. However, one question would be how did the
lipid class / fatty acid composition compare between the intact and the recombined
(skeleton+tissue) samples – were profiles affected? (Similar to the data presented for the proximate
composition presented in Table 1). Since the primary focus of the manuscript is on differences of
lipid extraction techniques it would helpful to have this data presented (can be presented in
supplementary tables as figures describe differences between skeleton and removed tissue).

Line 264-265: For the fatty acid composition, is it possible to state which of the SFA were dominant
(e.g. palmitic, oleic) as well as for PUFA and MUFA. I would also suggest that the major fatty acids
are presented in Supplementary Table 2 to save the reader time from looking through all the raw
data.

Discussion

The discussion is well structured and the authors use the literature well to discuss the results
obtained.

Figures

Would suggest selecting different colour shades used in the figures, perhaps one in white, as they
can be slightly difficult to distinguish when printed in greyscale.

Tables

Table 1: For consistency, please ensure all values are presented to 1 dp. For example, Organic for A.
millepora is 66.1 + 2.3, but Ash presented as 944 + 2.3. Thus, change to 944.0 + 2.3.

Were stats performed on the Crush Intact and Recombined values to show differences? If so, please
include lettering or asterix (*) to indicate significant differences. For example, if the lipid of A.
millepora for Crush Intact (9.9 + 0.6) and Recombined (6.2 + 0.6) are different, indicate using either
lettering or asterix (*) and indicate significance value. Similarly, stats between tissue and skelton.

Supplementary Table 1. The table shows the lipid class composition of the intact corals analysed.
Were any statistics performed between species, if so please provide. Authors also use Figure 2. to
show differences between the lipid class composition of skeleton and tissue. However, I was
wondering whether it was possible to provide a table showing the profile of intact and recombined
nubbins?

In addition, please ensure consistency when using decimal places (i.e. all values are presented to 1
dp, even if the value is a rounded number). Similarly, for the 0 value, you may want to include nd
(not detected).

Supplementary Table 2. Again, please ensure values are given to the same decimal places. Readers
may gain more information from the table if the major individual fatty acids are presented. For
example, for saturated fatty acids these may just be 14:0, 16:0, 18:0 – superscript lettering may be
included next to sum of totals with footnotes denoting which fatty acids are included, but not
presented within the table. This would save the reader from spending a fair amount of time sifting
through the attached raw data.

Were stats performed between species to show differences? If so, please indicate.

As for Supplementary Table 1 (Lipid Class data), is it possible to include values for recombined as
well as intact data?

---

## Round 0.2 · accepted · Accept

I see that you have responded to all the points raised by the reviewers and I confirm that you have made the necessary changes in the final submitted version.